# Study on the Roof Solar Heating Storage System of Traditional Residences in Southern Shaanxi, China

**DOI:** 10.3390/ijerph182312600

**Published:** 2021-11-29

**Authors:** Shuo Chen, Bart J. Dewancker, Simin Yang, Jing Mao, Jie Chen

**Affiliations:** 1Faculty of Environmental Engineering, The University of Kitakyushu, Kitakyushu 808-0135, Japan; bart@kitakyu-u.ac.jp (B.J.D.); yangsimin01@gmail.com (S.Y.); 2School of Electronic and Information Engineering, Ankang University, Ankang 725000, China; yuehediao@163.com (J.M.); chj526610398@126.com (J.C.)

**Keywords:** solar heating system, thermal storage roof pool, traditional residence, southern Shaanxi

## Abstract

Solar energy is a renewable, green, clean, and universal resource that has great potential in rural areas. Combining solar heating technology with building design to increase indoor thermal comfort in winter is an effective energy-saving and environmentally friendly approach. The factors affecting solar building heating mainly include two aspects; one is the lighting area of the building, and the other is the storage of building materials. By increasing the lighting area and using materials with good heat preservation and storage performance, the indoor temperature in winter can be effectively increased, and the heating time can be prolonged, thus decreasing the energy requirements of the building. In this paper, traditional houses in cold winter areas are selected as the research object, and a roof solar heating storage system is proposed. The method is to transform the opaque roof of the traditional houses into a transparent glass roof, and the thermal insulation and heat storage material HDPE is installed in the attic floorboards. The working principle of this system is to increase the amount of indoor solar radiation to raise the indoor temperature and make use of the thermal insulation performance of heat storage materials to prolong the indoor heating time. Through ANSYS software simulation, the heat transfer process, heat transfer mode, and temperature change of the system are analyzed, and the energy saving of the system is analyzed. The system can effectively raise the indoor temperature and has good energy-saving performance. The indoor temperature is raised by 5.8 °C, and the annual heat load of the building is reduced by 1361.92 kW·h, with a reduction rate of 25.02%.

## 1. Introduction

### 1.1. Motivation

Along with the fast growth of the economy, the consumption of energy in all walks of life globally has risen year by year. The world’s total energy consumption continued to grow from 1990 to 2018 [1]. In 2018, the world’s total energy consumption was approximately 9,937,702 ktoe, of which the total energy requirement of the construction industry was 2,839,313 ktoe, accounting for 28.77% of the total. It can be seen that the construction industry is a major force in energy consumption, and this phenomenon is particularly prominent in China [2]. In 2018, China’s total energy requirement was about 2,066,635 ktoe, while the total energy requirement of the construction industry was 997,672 ktoe, accounting for 48.28% of the total energy remand of various industries in China. The energy requirement of China’s construction industry is relatively high [3]. Based on current energy demand, it is expected that carbon emissions will continue to raise. The increase in energy consumption has led to the aggravation of energy shortages and environmental pollution in our country, becoming one of the most urgent problems in China [4]. 

In China, the current rural population is almost 737 million, accounting for 56% of the country’s entire population. The rural residential area is almost 23 million hm^2^, accounting for 60% of the country’s entire residential area, for which the rural residence energy demand accounts for 37% of the country’s total building energy requirements [5]. In 2018, rural residential carbon emissions were 497,422 ktoe, accounting for 37% of China’s construction industry carbon emissions, an increase of 13.23% compared to 2005 [6]. The demand for energy in rural residences is increasing year by year, and so energy-saving research on rural residential buildings is very important to China [7]. Facing the energy-saving problem of rural houses, many scholars proposed energy-saving designs through many research methods [8]. These can be roughly divided into the following aspects: the optimization of building materials, passive building design, and so on. These studies can reduce building energy consumption partly, however, a large number of experiments are still needed to prove the effectiveness of materials, economics, and whether it is a completely passive building [9]. 

### 1.2. Literature Studies

Regarding the research on building energy saving, Krarti et al. proposed the use of new energy to optimize the indoor temperature to save energy [10], including solar energy [11] and geothermal energy [12]. Applying passive solar energy technology to buildings is of great significance for energy-saving [13]. Chi et al. developed many energy-saving technologies in combination with solar energy, and the working principle is mainly to improve indoor temperature by using solar energy [14]. The main research content includes three aspects. The first is the use of phase change materials (PCMs) [15,16], the second is the partial renovation of buildings [17,18], and the third is architectural design strategies [19,20].

The first aspect of research is the use of PCMs, such as translucent photovoltaic roofs [21], Tromba walls [22], and thermal storage materials [23,24]. Cossu et al. [25] proposed combining silicon micro-cells with photovoltaic modules and setting them between glass plates, and then using them on sloping roofs. The results show that the design will not completely block the sun, and the shading rate is 9.7%. Spherical micro-batteries have been proven to be beneficial to improve the greenhouse system. Martins et al. [26] studied the improvement of a building’s energy-saving performance by integrating a Trombe wall system, which is another passive solar device. The ventilation system in a large volume is directly proportional to the thickness of the Trombe wall, and the Trombe wall is added in the building envelope. The addition of Trombe walls to the building envelope can reduce heating energy consumption by 16.36%. The device has a simple structure and zero working energy consumption and has been widely used in buildings. In order to decrease cooling energy consumption and carbon dioxide emissions, Irshad et al. [27] used double glazing and argon space on the Trombe wall, which proved to be feasible under the Malaysian climate. Shen et al. [28] studied the thermal performance of the traditional Trombe wall and the composite wall, established a mathematical model using the finite difference method, and verified it through experiments. Through the simulation of the software TRNSYS and FDM, the results showed that in cold and cloudy weather, the composite wall has better winter insulation performance than the traditional wall. Duan et al. [29] proposed that the thermal efficiency of the Trombe wall is related to the orientation of the building. Rabani et al. [30] analyzed a new Trombe wall that can absorb solar radiation in three directions and conducted research and analysis on its heating performance. Experiments showed that the model has good stability; the new Trombe wall can cause the indoor temperature to reach 10 °C. Al et al. [31] designed four types of vents for the glass roof through fluid mechanics and used Fluent to analyze the surrounding airflow field. The results showed that, in terms of wind speed and quality, vents perpendicular to the roof tilt angle provide the best performance. Wi et al. [32] objectively estimated the thermal performance of 21 commonly used insulating materials. The thermal conductivity was measured by the calorimeter method, and the water absorption was evaluated by a standard method, taking into account the drying conditions to ensure the sustainability of the material and long-term thermal conductivity. So, to solve the environmental pollution and environmental problems caused by plastics, Chavan et al. [33] added heat storage materials (TSM), such as functionalized graphene to recycled plastics, and evaluated the thermal characteristics, and found that the optimal concentration of reinforcing materials was added, meaning that the thermal storage capacity of the plastic increased by 54%. Mohammed et al. [34] studied a variety of PCMs and concluded that HDPE has high latent heat of PCMs, high phase change temperature, and is easy to use. Through experiments on the characteristics of various TSM, Song Jing et al. [35] analyzed and concluded that polyethylene has a long service life, stable performance, basically no supercooling and delamination, good mechanical properties, is easy to process and form, and has good practical application value. Xian et al. [36] studied composite PCMs such as ultra-high-molecular weight polyethylene and paraffin. Through microwave sintering, they had high energy storage efficiency in photothermal conversion. Kun [37] synthesized polyethylene glycol and rice husk ash into a new environmental protection material, which raised the heat storage potential by 20.9%. Piyachai et al. [38] studied polymer PCMs, designed the copolymerization of polyethylene glycol and ester polymers, developed the transition polymer PCD, and greatly increased the heat storage potential.

The second aspect of research is the partial renovation of buildings, such as solar chimneys, which are indoor heating devices used in buildings in winter [39]. Through the renovation of the roof, the glass chimney extends out of the roof at a certain inclination angle to receive more solar radiation [40]. The working principle of this equipment is to use passive air circulation to increase the heating effect. The heating principle is the same as the Trombe wall [41]. The research results show that solar chimneys in buildings can decrease the mean daily power requirement of air conditioning in winter by 10–20% [42,43]. Jing et al. [44] set the height ratio of different distances between the solar chimney and the roof, and the selected value range was 0.2–0.6. It was calculated by the model that when the height ratio of the optimal spacing is 0.5, the airflow velocity in the solar chimney is the largest. Miyazaki et al. [45] studied the performance of solar chimneys and programmed software to calculate the cooling and heating loads by CFD. Calculated by this program and compared with basic ventilation, the energy demand of the fan shaft was reduced by 50% throughout the year. In addition, the solar chimney reduced the heat load by about 20% and the annual heat load reduced by about 12%. 

The third aspect of research is the architectural design strategy, which mainly includes research on the building orientation [46,47] and the window-to-ground ratio [48], and the most suitable combination of orientation and window-to-ground ratio is obtained, so that the building can receive the maximum solar energy. The amount of radiation effectively increases the indoor temperature in winter to achieve the purpose of energy saving and emissions reduction. Attia [49], Marino [50], and others have studied the window-to-wall ratio (WWR) of buildings. So, to control and adjust the effect of solar radiation and airflow entering the room, a comprehensive analysis of airflow density, temperature, and wind speed was carried out, and the impact of building energy costs was considered. The research results show that WWR is less affected by the external environment and more affected by thermal properties [51]. Obrecht et al. [52] used quadratic equations to calculate the optimal WWR under different European climates, and they showed that the optimal WWR range is 38–42%. Ashrafian et al. [53] studied multiple groups of WWRs of a school building in Turkey and showed that, when the daylighting rate of glass is 50%, it can reduce the demand for artificial lighting and meet the indoor daylighting rate of 15%. Goia et al. [54] studied office buildings located in Italy based on energy demand standards. When the WWR is within the range of 35–45%, the energy demand is the smallest. Goia [55] used Energy Plus software to study office buildings in different climate zones in Europe, mainly analyzing indoor lighting and thermal performance to explore the best WWR. Although the optimal WWR differs due to the environment, orientation, and climate, when the WWR is between 30% and 45%, the energy requirement of the building is reduced by 5–25%. Wen et al. [56] used local climate and window orientation as the main indicator parameters to study the facades of office buildings in 10 different regions in Japan to evaluate the best WWR, and they also studied WWR and building carbon dioxide emissions.

The above research mainly analyzes the materials, local transformation, and design strategy and has made good progress and produced promising results. However, the effect on increasing indoor temperature is limited. In the research of PCMs, in addition to a large number of experiments, we also need to consider the production efficiency and economy. In terms of local reconstruction research, taking the research on glass roofs and solar chimneys as an example, scholars only consider changing the roof into glass to raise the range of solar radiation received in the building. However, with the gradual reduction in solar altitude angle, the indoor temperature also decreases, and the practice of absorbing and storing solar energy is not considered. Although the Trombe wall can reduce energy consumption, it only transforms the enclosed structure to decrease the heat transfer rate, so as to achieve the effect of thermal insulation. However, there is no research on the heat source. On this basis, this study proposes a new passive heating design—that is, the traditional residential roof solar heat storage heating system—which combines the building roof with renewable energy. Firstly, the glass skylight is installed in a traditional residential roof to increase the indoor solar radiation range, and the HDPE thermal insulation and heat storage material is added to the attic floorboards. By absorbing and storing solar thermal radiation, it can not only delay the reduction in indoor temperature and increase indoor heating time, but it can also reduce energy consumption, improve energy utilization, and reduce carbon emissions.

### 1.3. Scientific Originalities

Affected by the monsoon climate in southern Shaanxi, the indoor environment of traditional houses is generally wet and cold in winter. The existing heating methods are mainly grouped into two types: the traditional heating method is mainly heating by burning charcoal, and the modern heating method is mainly local heating with small electrical appliances. These two heating methods only heat part of the indoor space, and the energy consumption is high. This study proposes a roof solar heat storage heating system, which adds a roof skylight to the roof of traditional houses and adds the HDPE heat storage and insulation material to the roof attic plate, so that the polyethylene material on the attic plate can absorb the heat radiation of the sun and improve the overall indoor temperature by means of the heat transfer principle of heat convection and heat radiation. This heating system is different from the previous solar glass roof heating system. The roof attic plate is made of the solar panel heat storage material—high-density polyethylene (HDPE). This material has been proved to be a promising phase change material with many advantages, such as small volume change, high thermal efficiency, no corrosion, light undercooling, and so on [34,35]. For the purpose of increasing the efficiency of absorbing solar heat radiation, the traditional opaque roof in southern Shaanxi is transformed into a push-pull transparent glass roof. In winter, the thermal insulation roller shutter is closed to ensure that the solar panel heat storage material absorbs more heat radiation from solar radiation. In summer, opening the thermal insulation roller shutter can effectively reduce the indoor temperature. The whole heating process completely depends on passive solar heating without any power equipment support. It is a complete passive heating technology.

### 1.4. Aim of the Study

The thermal comfort of traditional residences in southern Shaanxi is poor in winter. The existing heating methods have some disadvantages, such as low efficiency, high energy consumption and high carbon. This study proposes a roof solar heating storage system to improve the indoor heating effect by means of the principle of solar heating technology. Chi et al. [14] confirmed that the ANSYS simulation results are similar to the measured data by comparison. Therefore, this study uses ANSYS software for simulation to verify the effectiveness of roof solar heating storage system heating. The research objectives are as follows:Measuring the heating time, equilibrium temperature, and heating efficiency of the basic system.Calculating the heat storage performance and heating efficiency of the attic slab material.Measuring the heating time, equilibrium temperature, and heating efficiency of the new system.Evaluating the appropriate times for using the new system for the entire year and calculating the annual heat load reduction and energy-saving effect when using the new system.

## 2. Methodology

### 2.1. Current Situation Research

#### 2.1.1. Research Location

Xianhekou village belongs to Ankang City, Shaanxi Province. It is a traditional village in the Qinba mountain area, covering an area of about 0.02534 km^2^. Affected by the mountain valley, the village developed in a belt shape along the transit road (Figure 1a). The village is in a group layout. Most of the residential buildings are 1–2-story traditional residential buildings, with a building density of about 19.64%. Therefore, this layout causes the residences to have good sunshine conditions. The proportion of one-story residences is 79.6%, and the proportion of two-story residences is 20.4%. Through field measurement, it was found that the average height of the one-story residences is about 3.5–4.5 m, and that of the two-story residential buildings is about 5.0–6.0 m. Southern Shaanxi is a region with hot summers and cold winters, and winters are wet and cold; according to China’s national meteorological data from 2015 to 2020, the lowest temperature in the village was −3 °C (Figure 1b). Coupled with the multiple influences of the residential building structure and building materials and other factors, the indoor temperature of the residence is lower in winter. The maximum and minimum wind speeds are 4.2 and 0.2 m/s throughout the year, respectively (Figure 1c).

#### 2.1.2. Research Method

This study mainly used the following research methods:

Field research: we conducted site surveys in Xianhekou Village and obtained first-hand information through surveying and photographing and by performing questionnaires, interviews, and other methods.

Data collection: data collection refers to the collation and analysis of field survey data, and, in this study, this process focused on the collection of data on building form, structure, house type, size, material, roof, attic area and size, door and window location and size, indoor and outdoor temperature, the use of heating equipment, and the length of use, etc.

Induction and summary: We classified and summarized the above data and selected representative traditional houses as the key research objects, summed up their commonalities and individualities, and finally chose traditional houses represented by house-style houses as the research objects.

#### 2.1.3. Research Content

Most of the traditional residences in southern Shaanxi have sloping roofs, one or two floors of a civil or brick concrete structure, and the second floor is low in height. It is mainly used as a storage room. Due to the cold winter in southern Shaanxi, the indoor temperature is low. However, due to the high and uneconomical energy consumption of air conditioning and other power equipment, it is not popularized locally, and so passive heating is a good choice. Combined with the characteristics of traditional residences in southern Shaanxi, local transformation is carried out. A glass skylight is set on the roof, a push-pull thermal insulation roller shutter is added above the skylight, and thermal insulation and heat storage materials are added on the attic floor. In winter, the glass skylight is opened to allow the heat storage materials to fully absorb the solar radiation. The thermal insulation and heat storage performance of TSM was used to improve the indoor temperature. A local traditional residence was selected as the research object. The studied building is one floor as a whole and two floors locally. The sloped roof form is adopted. The story height of the first floor is 3 m and the second is 2.8 m. The wall of the residence is made of red bricks, and the sloped roof is made of green slate or tiles. The model is established in this form (Figure 2a). Southern Shaanxi is wet and cold in winter. In addition, the residential houses are closely arranged and have long cornices, which reduces the solar radiation obtained by the residential houses in winter. Therefore, it is very wet and cold indoors. To solve this problem, local residents adopt active and passive heating methods. Active heating includes small suns, electric stoves, and electric blankets. Passive heating includes fire pots, charcoal fire basins, etc. The above two heating methods aim to improve the temperature of a certain part of the room, rather than the overall indoor temperature. Through the research on the roof solar heating storage system, it is concluded that the key to system design lies in the size selection and practice of installing roof transparent glass and the selection of heat storage materials as the attic floor. By comparing the physical properties of HDPE with other thermal insulation and heat storage materials, it is concluded that HDPE has a higher phase transition temperature and latent heat (Table 1) [34]. Therefore, HDPE was selected as heat storage material in this study. The core components of the solar heating system on the roof include heat storage attic floorboards and a roof skylight. The design method of the heat storage attic floorboards was placing a 5 cm-thick HDPE board on the attic floor, so as to change the ordinary attic floor into a heat storage attic floor and improve the thermal insulation and heat storage performance (the red area in Figure 2b). The design method of the roof skylight was to design three groups of 3 m × 3 m ordinary roof opening fans and glass skylights on the south-facing sloped roof, using 6 mm-thick single-layer glass with a shading coefficient of 0.92 and a heat transfer coefficient of 5.818 W/m^2^k (the blue area in Figure 2b). In addition, considering the influence of thermal insulation and heat storage materials on the indoor environment in summer, a thermal insulation roller shutter was set on the roof ridge (the yellow area in Figure 2b). In winter, the solar energy heating system is started, the thermal insulation roller shutter on the glass is closed, and the heating system starts to work to make the heat storage attic floor absorb solar radiation (Figure 2d). In summer, the thermal insulation roller shutter on the glass (Figure 2e) can be opened to effectively reduce the impact of solar radiation on the room. Therefore, the opening and closing of the thermal insulation framework can effectively improve the indoor temperature in different seasons. The working principle is that the roof solar heating storage system uses solar heat radiation to heat the thermal storage attic plate through the roof glass skylight, and it uses thermal convection and radiation to affect the temperature between the walls, so as to improve the temperature of the whole room (Figure 2c).

#### 2.1.4. Comparative Analysis of Current Heating Methods

Through the field investigation of heating methods, it can be seen that there are mainly two existing heating methods, namely active heating and passive heating. In southern Shaanxi, active heating methods mainly include small suns, electric stoves, and electric blankets. These methods’ disadvantages include high energy consumption and being uneconomical. According to the investigation, the main parameters and applications of these methods are as follows: when a small sun is used, the power is either between 300 and 500 W, accounting for 36.94% of instances; between 500 and 800 W, accounting for 53.68% of instances; between 800 and 1200 W, accounting for 9.38% of instances. In addition to the power required in use, the time in use should also be considered. In a day, the use of a small sun for 0–3 h accounts for 32.65% of instances; the proportion of using a small sun for 3–6 h is 42.66% of instances; the proportion of using a small sun for 6–9 h is 24.69% of instances. It can be seen that the use frequency of small suns is high, and the time spent using small suns is long.

The frequency of electric furnace use is low because of their high power and large power consumption. Through field measurement, it is found that the power of the electric furnace is generally 2400 W, and it is operated for 5 min, which is equivalent to the consumption of 0.2 kW·h. The use frequency of electric blankets is the highest, because the use time of an electric blanket is generally at night, which mainly improves the local indoor temperature. Through the investigation of the power and service time of the electric blanket, it can be concluded that when the electric blanket is used, the power is either between 60 and 80 W, accounting for 31.82% of instances; the power is between 80 and 100 W, accounting for 39.39% of instances; or the power is between 100 and 120 W, accounting for 28.79% of instances. In a day, the proportion of using an electric blanket for 0–1 h is 45.65%; the proportion of using an electric blanket for 2–3 h is 32.36%; the proportion of using electric blanket for 3–4 h is 21.99% of instances. It can be seen that the three active heating methods mainly rely on electric energy, and the heating effect is poor, as these methods can only heat parts of the indoor space (Figure 3).

Passive heating mainly includes charcoal pots and stoves, mainly burning firewood. The proportion of households using charcoal fire pots was 45.68%, and the proportion of households using stoves was 54.32%. Because passive heating methods have an open heat source, they will emit more carbon dioxide during heating, which is not conducive to environmental protection. Whether it is the small sun, electric stove, and electric blanket for active heating or the carbon fire basin and stove for passive heating, the thermal efficiency is poor, and only part of the indoor space is heated.

### 2.2. Roof Solar Heating Storage System

#### 2.2.1. Design of the Roof Solar Heating Storage System

According to the survey, most of the traditional residences in southern Shaanxi have one story and are L-shaped, and the space generally has three bays. The roof is mostly made of tiles and slate, and it is a sloped roof; the wall is an adobe wall plus lime painting. Due to the use of sloping roofs, residences have good daylighting performance. In addition, the attic is usually set on the top, which provides an important basis for the design of the roof solar heating storage system. To make the simulation true and reliable, a dwelling with three bays, and one deep sloping roof was selected as the model. The design of the roof solar energy storage heating system includes two parts. The first part is the roof daylighting area. A glass skylight is designed on the south side of the sloping roof, which is made of silica—the transmittance is 0.87, the thickness is 6 mm, and the size is 3 m × 3 m. As the residence is divided into three bays, a glass skylight is set in each bay. The second part is the setting of the attic floor. HDPE heat storage material with a thickness of 5 cm is installed in the attic floor, which is also the core of the system. The working principle of this system is that the heat storage material absorbs and stores the heat radiation of the sun during the day, and the exothermic performance of TSM is used to improve the indoor temperature through the heat transfer form of heat radiation and heat convection at night.

#### 2.2.2. Working Principle of the Roof Solar Heating Storage System

The working principle of the solar heat storage system is to heat the attic floor panel through the roof skylight by solar radiation and use the thermal insulation and heat storage performance of HDPE to improve the indoor temperature in the form of thermal radiation and convection. The roof solar heating storage system can be regarded as a solar energy recycling device. The working process of the system is affected by many factors, including the thermal properties of heat storage materials, glass thickness, room size, and indoor and outdoor temperature. The thermal transfer process has three parts: the first part is the heating of HDPE through glass by means of solar thermal radiation. This heat transfer process is a form of thermal transfer dominated by thermal convection and thermal radiation. The thermal transfer time is mainly based on transient–steady–transient heat transfer, which belongs to the coupled heat transfer of solids and fluids. According to Planck’s law, the calculation formula is as follows (Equation (1)):(1)$$\mathrm{Eb}\lambda =\frac{C_{1}\lambda^{-5}}{E^{C_{2}/\left(\lambda T\right)}-1}$$
where Ebλ is the blackbody spectral radiant power, which is the wavelength, T is the blackbody thermodynamic temperature, E is the base of the basic logarithm, C_1_ is 3.7419 × 10^−16^ W·m^2^, and C_2_ is 1.4388 × 10^−2^ m·k.

The second part is the heat storage of the HDPE material itself. This heat transfer process is a form of heat transfer dominated by heat conduction. The heat transfer time is mainly transient–steady–transient heat transfer. It belongs to the solid–solid coupling heat transfer. According to Borlier’s law, the calculation formula is as follows (Equation (2)):(2)q=−λgradt=−λn
where q is the space heat flux vector, λ is the proportional coefficient, gradt is the temperature gradient at a certain point in space, the temperature change rate of the object along the x direction, and n is the normal unit of the isotherm passing through the point.

The third part is when the HDPE material reaches a constant temperature—the temperature between the walls is heated to raise the temperature of the entire room through heat transfer, mainly in the form of heat convection and heat radiation. This process belongs to the coupled heat transfer of solids and fluids, and the heat transfer time is mainly based on transient–steady–transient heat transfer. This follows Newton’s law of cooling (Equation (3)):(3)$$q=h\left(T_{\mathrm{body}}\;-\;T_{\infty }\right)=h\Delta T$$
where q is the heat flux density, h is the material convection heat transfer coefficient, T_body_ is the temperature of the object, T_∞_ is the temperature of the surrounding environment, and T is the difference between the surface and the ambient temperature. When the sun’s thermal radiation disappears, the heating of the HDPE heat preservation and heat storage material will stop.

#### 2.2.3. Analysis of Basic Heating

(1)With solar radiation

The winter in southern Shaanxi is relatively cold, and the commonly used heating methods include active and passive heating methods. Among them, passive heating is mainly basic heating, which mainly includes two aspects: on the one hand, it relies on solar heat radiation, and on the other hand, it relies on indoor heat distribution sources. Basic heating is the combined effect of solar thermal radiation and indoor thermal distribution sources with solar radiation. Solar thermal radiation mainly means that the outer wall of the wall relies on solar thermal radiation to heat the room to raise the temperature of the room. The heat transfer process is as follows: the first step is the absorption of heat by the outer surface of the wall, which mainly absorbs solar energy through convective heat transfer and radiation. The second step is the heat transfer of the wall itself, and the heat is transferred from the outer wall to the interior wall. The third step is the internal wall radiating heat. After the temperature of the inner wall rises, the indoor temperature rises through the combined effect of heat conduction, convection, and radiation. The heating efficiency of basic heating is low, and the increase in indoor temperature is small. Once the indoor temperature is constant, it is only 2.5 °C higher than the outdoor temperature.

(2)Without solar radiation

Basic heating mainly relies on indoor heat sources, mainly including heat generated by electrical appliances, cooking and the human body. The change in interior temperature will also be affected by building size, material, and structure. In the previous heating research, the two factors of solar thermal radiation and indoor heat distribution sources were considered less often, which made the simulation results poor. In this study, the comparison between the new system and basic heating can fully illustrate the practicability and authenticity of the new system. The interior temperature is only affected by the heat source without solar radiation, and the range of interior temperatures changes to a lesser degree, only rising by 0.1 °C compared with the outdoor temperature.

#### 2.2.4. Analysis of the Situation of the Roof Solar Heating Storage System

(1)With solar radiation

The roof solar energy storage system is used to absorb solar energy to improve the indoor temperature of the residences, and the heat absorption and storage capacity of the heat storage plate is used to raise the indoor temperature to achieve the effect of indoor heating in the winter. The heat storage plate adopts HDPE material with good heat absorption and heat storage effects, which can store and release heat while absorbing heat. The working principle of the process of the pool solar heating system absorbing, storing, and releasing heat is shown in Figure 2c. When the heat storage plate absorbs heat, the temperature of the heat storage plate increases significantly. After the temperature of the heat storage plate increased for 1 h, the indoor temperature gradually increased, rising by 0.7 °C. The bedroom temperature reached the highest value of 8.9 °C at 16:00. Compared with basic heating, the bedroom temperature increased more. The heating effect of residential buildings under the new system is better, and the temperature and time spent after heating improved.

(2)Without solar radiation

Considering that without solar heat radiation, only the effects of indoor heat distribution sources and electrical appliances are considered, the temperature of the heat storage plate rises slowly and is not much different from the indoor and outdoor temperatures. It can be seen through the simulation of ANSYS software (American ANSYS company, Pittsburgh, PA, USA) that the indoor temperature increased by 0.3 °C. In previous studies, less consideration was given to the effects of indoor heat distribution sources and electrical appliances. The addition of these two factors this time can more fully illustrate the effectiveness of the new system.

### 2.3. Simulation Analysis

The simulation software includes ANSYS, Rhino (Robert McNeel & Assoc, Seattle, WA, USA), EnergyPlus (National renewable energy laboratory, Golden, CO, USA), and Matlab (MathWorks, Natick, MA, USA), and the boundary conditions are the presence and absence of solar heat radiation on the wall. ANSYS mainly computes the time required for room heating under basic heating, the new system, and the final temperature. Rhino software is mainly used to build models, and the reduction in heat load is calculated by Ladybug and EnergyPlus. Matlab is used to program the analysis results to output the calculation results. The software ANSYS is mainly used to simulate and calculate steady-state, transient thermal power, and thermal radiation. The steady-state and the transient thermal power are used to compute the heat transfer between the heat storage plates. Thermal radiation simulation mainly computes the solar thermal radiation and the change in interior temperature after the heat storage plate released heat. EnergyPlus computes the heat load after adopting the new system. This process includes four steps: building a model in the rhino, importing the model into the ladybug to set relevant parameters, using Energyplus to calculate the heat load circumstances, and finally, using Matlab software to program and output the calculation results.

The simulation in this study included three steps. The first step was to simulate basic heating by building a model based on the measured data and using local winter weather conditions data. The main content was the temperature change after heating. The second step was to simulate the absorption and heat release of the heat storage board after establishing the roof solar heating system, which mainly included a sloping roof skylight and a heat storage board. The content of this simulation was the temperature change of the heat storage plate, the heating time of the outer wall of the heat storage plate, the heating time of the inner wall of the heat storage plate, and the time required for the indoor temperature after equilibrium. The third step was to simulate the suitable time and the reduction in heat load.

## 3. Results and Discussion

### 3.1. Comparison between the Basic Heating System and the New System without Solar Heat Radiation

#### 3.1.1. Basic Heating without Solar Heat Radiation

To verify that the new system is still effective without solar heat radiation, the comparative analysis was based on basic heating without solar heat radiation. Basic heating without solar radiation mainly relies on interior thermal distribution to change the indoor temperature. Using software simulations, indoor temperatures under basic heating are similar to outdoor temperatures without solar radiation (Figure 4).

The building materials of traditional houses in southern Shaanxi include red brick, gravel, concrete, and wood. The simulation steps are as follows: first, using Rhino software to build a residential model, then importing the model into ANSYS software, and setting the grid, activating the energy equation, selecting the turbulence model, setting the radiation model, adding the thermal properties of the above materials, and assigning materials to the model properties. Then, the boundary conditions and solution method are set. Finally, the flow field to initialize the solution simulation process and performing simulation calculations is set. So, to compare the temperature change with solar thermal radiation, the simulation calculation time was 24 h, divided into day and night. The indoor temperature during the day was greatly affected by solar radiation, and the time was from 8:00 to 18:00. The indoor temperature at night was not affected by solar radiation, and the time was from 18:00 to 8:00. The following three heating conditions used this calculation time. The indoor temperature changed as follows: 18:00 to 8:00 was the night, and the indoor temperature was 2.6–3.8 °C. From 8:00 to 12:00 (Figure 5a,b), the mean temperature of the indoor room was 3.2 °C. The outdoor temperature was low at this time, which had little effect on the indoor temperature. From 12:00 to 14:00 (Figure 5c,d), the indoor temperature remained basically unchanged, and the mean temperature was also 3.3 °C. From 14:00 to 17:00 (Figure 5e), the outdoor and the indoor temperatures also began to rise slowly. After equilibrium, the temperature was 4.2 °C. From 17:00 to 18:00 (Figure 5f), the outdoor temperature dropped, and the indoor temperature was 3.6 °C. The mean indoor temperature was 3.7 °C without solar radiation, which was 0.5 °C higher than the mean temperature outdoors.

#### 3.1.2. The Roof Solar Heating Storage System without Solar Thermal Radiation

The new system without solar thermal radiation also depends on indoor thermal distribution to change indoor temperature. The indoor and outdoor temperature is balanced, which is the same as basic heating without solar radiation.

The new system is improved on the basis of the basic heating system, adding roof glass skylights and heat storage material attic slabs. The other building materials are the same as those used in the building with a basic heating system. The same method as in the basic heating experiment was used to build models, set material parameters, and simulate conditions. The indoor temperature changed as follows: from 18:00 to 8:00, the mean temperature of the attic floor was 2.7–3.9 °C. The indoor and outdoor temperatures were basically the same, at 2.7–4.0 °C. From 8:00 to 12:00, the mean temperature of the attic floor was 3.7 °C, and the mean temperature of the indoor room was 3.6 °C (Figure 6a,b). This had little effect on the indoor temperature. From 12:00 to 14:00, the mean temperature of the attic floor was 3.7 °C, and the indoor temperature remained basically unchanged, with a mean temperature of 3.6 °C (Figure 6c,d). From 14:00 to 17:00, the outdoor temperature rose slowly. Due to the use of electrical appliances and heating equipment during this period, the indoor temperature began to rise slowly. After equilibrium, the mean temperature of the attic slab was 4.4 °C, and the mean indoor temperature was 4.5 °C (Figure 6e). From 17:00 to 18:00, the outdoor and indoor temperatures decreased. The mean temperature of the attic slab was 3.5 °C, and the mean indoor temperature was 3.6 °C (Figure 6f). The mean indoor temperature did not change much, and it was greatly influenced by the outdoor temperature without solar radiation. The mean temperature of the attic floor was 3.9 °C, and the mean indoor temperature was 3.8 °C. The temperature variation range was small, and the difference in the average temperature between the indoors and outdoors was 0.6 °C.

#### 3.1.3. Comparative Analysis

The indoor temperatures of the buildings with basic heating and the roof solar heating storage system were the same without solar radiation. The interior temperature could be increased by 1.1 °C and the heating efficiency was 0.12 °C/h. Using the new system, the temperature could be increased by 1.4 °C, and the heating efficiency was 0.16 °C/h. The reason for this is that without solar radiation, the interior temperature was influenced by the outdoor temperature and indoor thermal distribution. Due to the cold outdoor weather and low temperature in winter, the indoor temperature was low. Therefore, without solar radiation, the heating efficiency of these two heating methods was low, and the improvement effect on indoor temperature was poor. 

### 3.2. Comparison between the Basic Heating System and the New System with Solar Heat Radiation

To verify the effectiveness of the roof solar heating storage system, the comparison was made based on the basic heating effect. The specific simulation methods are as follows: first, the residential model was established with Rhino software, then the model was imported into ANSYS software, and the grid was set, the energy equation was activated, the turbulence model was selected, the radiation model was set, the thermal properties of the above materials were added and the material properties were assigned to the model, and then the boundary conditions and solution methods were set. Finally, the flow field initialization solution simulation process was set up and the simulation calculation was carried out.

#### 3.2.1. Basic Heating with Solar Heat Radiation

Basic heating depends on solar radiation to increase the indoor temperature by irradiating the wall, glass, and roof with solar heat radiation. The simulation process and simulation time were consistent with those without solar radiation. According to the simulation calculation, from 8:00 to 12:00, the temperature of the east room and the central reception hall began to raise from the initial mean temperature of 3.1 to 5.6 °C. The west room was less affected by solar radiation, the temperature rose relatively slowly, and its temperature variation range was the same as that of the east room (Figure 7a,b). From 12:00 to 14:00, the indoor temperature changed slightly and became nearly stable, and the mean indoor temperature was 5.6 °C (Figure 7c,d). At 16:30, the maximum temperature of the west room reached 5.7 °C, and then it began to decline. The interior temperature in the east room decreased, and the room temperature began to drop from 5.6 to 3.8 °C (Figure 7e). From 17:00 to 18:00, the indoor temperature continued to drop, akin to the outdoor temperature, and the temperature was 3.2 °C (Figure 7f). Under the condition of solar radiation, the mean indoor temperature was 5.6 °C, which was 2.5 °C higher than the mean outdoor temperature.

#### 3.2.2. Roof Solar Heating Storage System with Solar Thermal Radiation

To compare the simulation results with basic heating, the outdoor temperature distribution is similar to that of basic heating. The indoor original temperature was also 3.1 °C. The heating process of the new system includes three stages: solar radiation heating the polyethylene attic floor, the heat transfer of attic floor itself, and indoor heating by the attic floor.

In the first stage, the polyethylene loft floorboard was heated by solar radiation, and the temperature at the top of the loft floorboard reached a stable state. Due to the change in solar height angle, the temperature changes of the eastern, central, and western loft boards were different, and the indoor temperature change in this stage was small. At 8:00 in the morning, the sun shined on the top of the polyethylene attic floor through the roof glass. The temperature of the east room rose first, followed by the central and western attic floorboards. At 11:40, the temperature of the attic floor was stable. Due to the short heating time of the attic floor in the morning, the indoor temperature changed little. The temperature changes are as follows:

At 8:00, the floor temperature was the same as the indoor temperature, which was 3.1 °C. At 9:00, the temperature of the eastern attic floorboard was 7.1 °C, the temperature of the middle attic floorboard was 6.2 °C, and the temperature of the western attic floorboard was 5.2 °C (Figure 8a). At 9:30, the temperature of the eastern attic floorboard was 7.9 °C, the temperature of the middle attic floorboard was 7.4 °C, and the temperature of the western attic floorboard was 6.9 °C (Figure 8b). At 10:00, the temperature of the eastern attic floorboard was 10.2 °C, the temperature of the middle attic floorboard was 8.6 °C, and the temperature of the western attic floorboard was 7.8 °C (Figure 8c). At 10:30, the temperature of the eastern attic floorboard was 16.5 °C, the temperature of the middle attic floorboard was 10.6 °C, and the temperature of the west attic floorboard was 10.3 °C (Figure 8d). At 11:00, the temperature of the eastern attic floorboard was 20.3 °C, the temperature of the middle attic was 19.4 °C, and the temperature of the western attic floorboard was 18.1 °C (Figure 8e). At 11:40, the temperature of the attic floor was nearly stable. The temperature of the attic floor in the east was 24.5 °C, the temperature of the attic floor in the middle was 24.3 °C, and the temperature of the attic floor in the west was 24.5 °C (Figure 8f).

The second stage was the heat transfer of the attic floor itself. During the heating process, the temperature at the top of the attic floor rose first, and then the bottom was heated. After simulation calculation, after 22 min, the temperature at the bottom of the attic floor was about 23.6 °C. The indoor temperature also began to rise slowly. The temperature change process at the bottom of the attic floor was as follows: at 11:45, the mean temperature of the attic floor was 10.6 °C (Figure 9a). At 11:50, the mean temperature of the attic floor was 15.5 °C (Figure 9b). At 11:55, the mean temperature of the attic floor was 19.4 °C (Figure 9c). At 12:00, the mean temperature of the attic floor was 23.7 °C (Figure 9d).

The third stage was the attic floor heating the room. The principle of this process was using thermal radiation to heat the indoor air to raise the indoor temperature. From 13:00 to 16:00, the attic floor temperature remained unchanged, and the indoor temperature continued to rise; at 17:00–19:00, the temperature of the attic floor began to decrease, and the indoor temperature also decreased. The changes in attic floor and indoor temperature are as follows:

From 13:00 to 16:00 (Figure 10a–d), the temperature of the attic floor remained unchanged at 23.5 °C. At 17:00, the temperature of the eastern attic floorboard was 21.4 °C, the temperature of the middle attic floorboard was 22.5 °C, and the temperature of the western attic was 7.6 °C (Figure 10e). At 17:30, the temperature of the eastern attic floorboard was 15.4 °C, the temperature of the middle attic floorboard was 16.7 °C, and the temperature of the western attic floorboard was 17.2 °C (Figure 10f). At 18:00, the temperature of the eastern attic floorboard was 10.6 °C, the temperature of the middle attic floorboard was 11.2 °C, and the temperature of the western attic floorboard was 12.4 °C (Figure 10g). At 19:00, the temperature of the eastern attic floorboard was 8.8 °C, the temperature of the middle attic floorboard was 9.1 °C, and the temperature of the western attic floorboard was 9.2 °C (Figure 10h). From 19:30 to 8:00, the temperature of the attic floor remained stable at 3.1 °C.

At 13:00, the temperature of the east room was 5.5 °C, the temperature of the middle room was 5.8 °C, and the temperature of the west room was 6.2 °C (Figure 10a). At 14:00, the temperature of the east room was 5.8 °C, the temperature of the middle room was 6.2 °C, and the temperature of the west room was 6.7 °C (Figure 10b). At 15:00, the temperature of the east room was 7.7 °C, the temperature of the middle room was 7.8 °C, and the temperature of the west room was 8.1 °C (Figure 10c). At 16:00, the temperature of the east room was 7.9 °C, the temperature of the middle room was 8.0 °C, and the temperature of the west room was 8.2 °C (Figure 10d). At 17:00, the temperature of the east room was 7.3 °C, the temperature of the middle room was 7.4 °C, and the temperature of the west room was 7.6 °C (Figure 10e). At 17:30, the temperature of the east room was 6.5 °C, the temperature of the middle room was 6.7 °C, and the temperature of the west room was 6.7 °C (Figure 10f). At 18:00, the temperature of the east room was 5.1 °C, the temperature of the middle room was 5.4 °C, and the temperature of the west room was 5.3 °C (Figure 10g). At 19:00, the temperature of the east room was 4.2 °C, the temperature of the middle room was 4.3 °C, and the temperature of the west room was 4.5 °C (Figure 10h). The indoor temperature remained stable at 3.1 °C from 19:30 to 8:00 the next morning. 

#### 3.2.3. Comparative Analysis

The reasonable use of solar energy can greatly improve the indoor temperature. The roof solar heating storage system was mainly affected by solar radiation with solar radiation. The initial temperature of the attic floor and the indoor space was 3.1 °C, and when the indoor temperature began to rise at 9:00, the mean temperature was 3.8 °C. At 16:00, the indoor temperature was 8.9 °C. The indoor temperature decreased at 17:00, and the mean indoor temperature was 7.9 °C. Therefore, the temperature was increased by 5.8 °C, and the mean heating efficiency was 0.97 °C/h. The basic heating system mainly relies on the heat generated by solar radiation on the wall to influence the indoor temperature. The temperature was increased by 2.5 °C, and the mean heating efficiency was 0.42 °C/h. Therefore, the new system mainly depended on the roof glass and the heat storage material attic floor to improve the indoor temperature. Compared with the basic heating system, it greatly increased heating efficiency and heating time.

### 3.3. Heating Time with the New System

The heat load reduced by means of the use of the roof solar heating storage system was estimated with or without solar radiation for the entire year. The time spent using solar heating was mainly distributed in January, February, March, November, and December. The solar energy system was used for 807 h during the year without solar radiation, accounting for 9.16% of the whole time (Figure 11a). The time spent using the new system in the whole year was 1141 h with solar heat radiation, accounting for 12.95% of the total time over the year (Figure 11b).

### 3.4. Energy-Saving Situation of Using the New System

The annual heat load of the new system was simulated by EnergyPlus—it was reduced by 517.84 kW·h without solar radiation, and the reduction ratio was 9.51% (Figure 12a), while it was reduced by 1361.92 kW·h with solar radiation, and the reduction ratio was 25.02% (Figure 12b). 

### 3.5. Thermal Comfort with the New System

Taking 18 °C as the limit of indoor thermal comfort, we calculated the time when the new system produced temperatures below 18 °C. Under the condition of basic heating, the time when the temperature of the building was lower than 18 °C was 5019 h (Figure 13a), accounting for 56.98% of the year. When the solar energy system was adopted, the time when the interior temperature was lower than 18 °C is 4006 h without solar radiation, accounting for 46.36% of the year (Figure 13b). The time when the temperature was lower than 18 °C all year was 3763 h with solar radiation, accounting for 43.55% of the year (Figure 13c).

### 3.6. Empirical Research

Nowadays, solar heating technology has been tried in rural houses in Ankang City, China. The research team conducted a preliminary study and experiment on the solar heating system on 6 January 2019. The experimental site was Lihuaping village, Houliu Town, Ziyang County, Ankang City. The experimental object was a one-story sloped-roof residential house. It faces due south, the wall material is 240 mm-thick red brick, and the roof is glazed tile, which is easy to transform. Its functions include a living room, bedroom, kitchen, and storage room (as shown in the figure). According to the demands of the heating system, only the living room and two bedrooms were heated, with a heating area of 66 m^2^. According to the requirements of heat storage roof heating system, the roof was transformed into a glass skylight, and a heat insulation board was placed below the skylight. The specific heat capacity of pebbles was C = 780 j/(kg·K), and the thermal diffusivity was a = 11.3 × 10^−7^ m^2^/s, which is more suitable as a heat storage material, and there are many pebbles in the local area. Therefore, 10 cm thick pebbles were placed on the attic floor as heat storage material. Next, the indoor temperature of the reconstructed residential house was measured. The test weather was sunny, the time period was 8:00–18:00 in the daytime, and the time interval was one hour. The changes in indoor temperature when the heat insulation board was closed and opened were measured, respectively, for one month. Additionally, effective temperature changes for sorting were selected (Table 2). Through analysis and comparison, the average indoor temperature was 6.2 °C in the ten days after closing the heat insulation board, 3.8 °C in the ten days after opening the heat insulation board, and the indoor temperature increased by 2.4 °C. Polyethylene was selected as the heat storage material in this study. Its specific heat capacity and thermal diffusivity are higher than pebbles, so the temperature increase range was greater than before, which also confirms the practicability of the system for improving the room temperature in winter.

## 4. Conclusions and Outlook

### 4.1. Conclusions

Traditional residences in southern Shaanxi are the research objects of this study. To raise the indoor temperature, the roof solar heating storage system proposed. First, we establish the roof solar heating storage system and changed the roof to a partial glass roof, so that the attic slab received the heat radiation from the sun. So, to improve the heat storage effect of the attic floor, HDPE was installed in the attic floor. At the same time, the heat storage material polyethylene was added to the attic floor. Second, the heating time and temperature change of the basic heating system and new system were calculated and simulated by ANSYS software. Finally, the results of the influence of the two heating methods on the indoor temperature were compared. The roof solar heating storage system can raise the indoor temperature, and the following conclusions were drawn:Using the thermal storage roof pool solar heating system, the indoor temperature increased by 4.9 °C with solar radiation, and the mean heating efficiency was 0.82 °C/h. The indoor temperature increased by 1.4 °C without solar radiation, and the mean heating efficiency was 0.16 °C/h.Using basic heating, the interior temperature increased by 2.5 °C with solar radiation, and the mean heating efficiency was 0.42 °C/h. The indoor temperature increased by 1.1 °C without solar radiation, and the mean heating efficiency was 0.12 °C/h.The time spent using the solar heating system was mainly distributed in in January, February, March, November, and December. The solar energy system was used for 807 h all year without solar radiation, accounting for 9.16% of whole year. The time spent using the new system in the whole year was 1141 h with solar heat radiation, accounting for 12.95% of whole year.The annual heat load of the new system was reduced by 517.84 kW·h without solar radiation, and the reduction ratio was 9.51%, while it was reduced by 1361.92 kW·h with solar radiation, and the reduction ratio is 25.02%.Taking 18 °C as the limit of indoor thermal comfort, we calculated the time when the new system was below 18°C. Under the condition of basic heating, the time when the temperature of the building was lower than 18 °C was 5019 h, accounting for 56.98% of the year. When the solar energy system was adopted, the time when the interior temperature was lower than 18 °C is 4006 h without solar radiation, accounting for 46.36% of the year. The time when the temperature was lower than 18 °C all year was 3763 h with solar radiation, accounting for 43.55% of the year.

### 4.2. Outlook

Future studies mainly include the following:The appropriate value of the roof transparent area ratio and the roof inclination angle can be further explored to obtain more solar radiation indoors and to achieve the best indoor temperature.The heat storage plate used HDPE heat storage material. In future research, we should combine the development of science and technology to explore more suitable heat storage materials to further increase the indoor temperature in winter.

## Figures and Tables

**Figure 1 ijerph-18-12600-f001:**
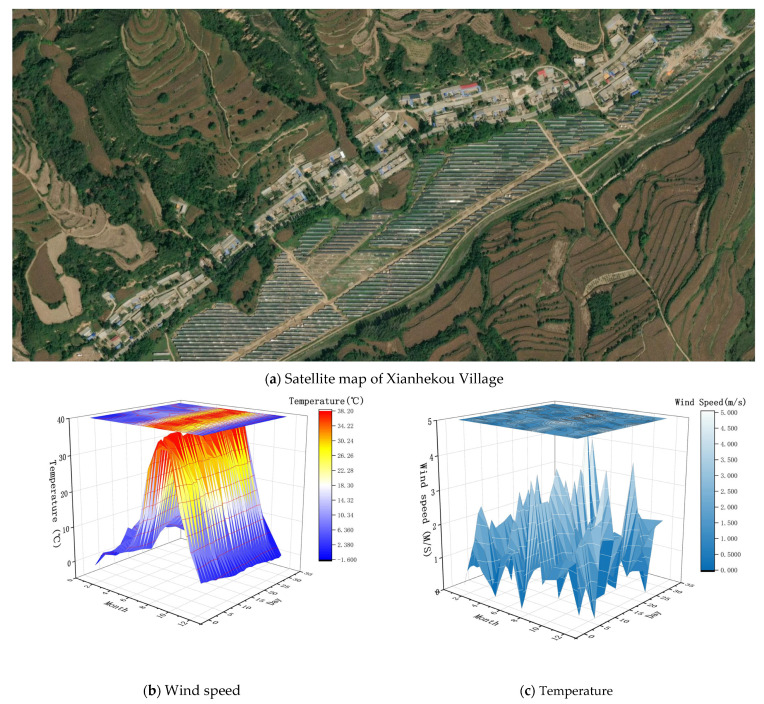
(**a**) Satellite map of Xianhekou Village; (**b**) wind speed chart of Xianhekou Village; (**c**) temperature distribution chart of Xianhekou Village.

**Figure 2 ijerph-18-12600-f002:**
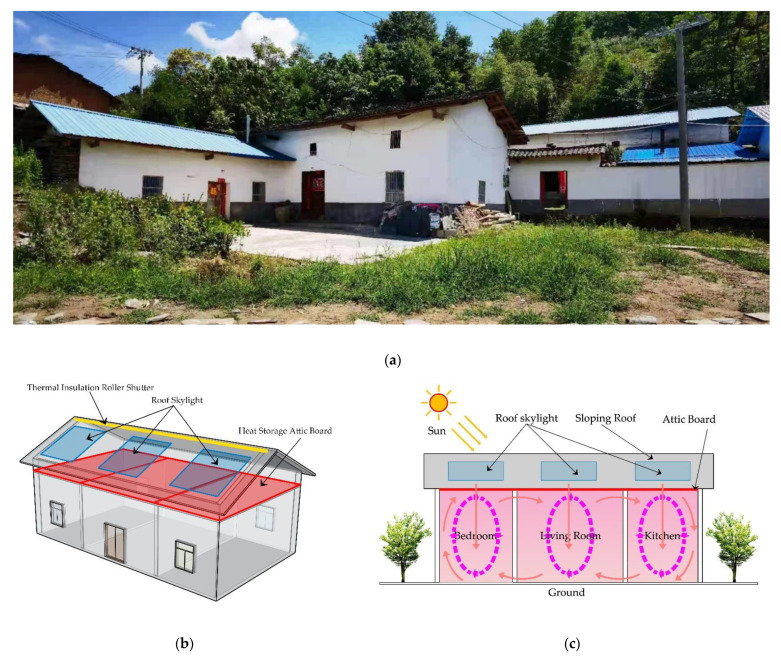
(**a**) Residential photos; (**b**) roof solar heating system; (**c**) working principle; (**d**) closing the thermal insulation roller shutter in winter; (**e**) opening the thermal insulation roller shutter in summer.

**Figure 3 ijerph-18-12600-f003:**
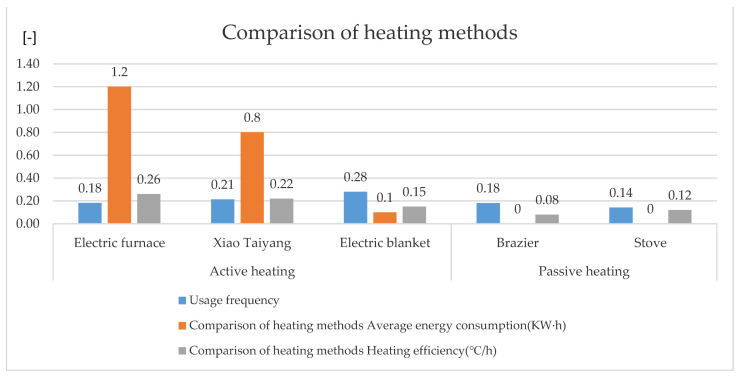
Comparative analysis of current heating methods.

**Figure 4 ijerph-18-12600-f004:**
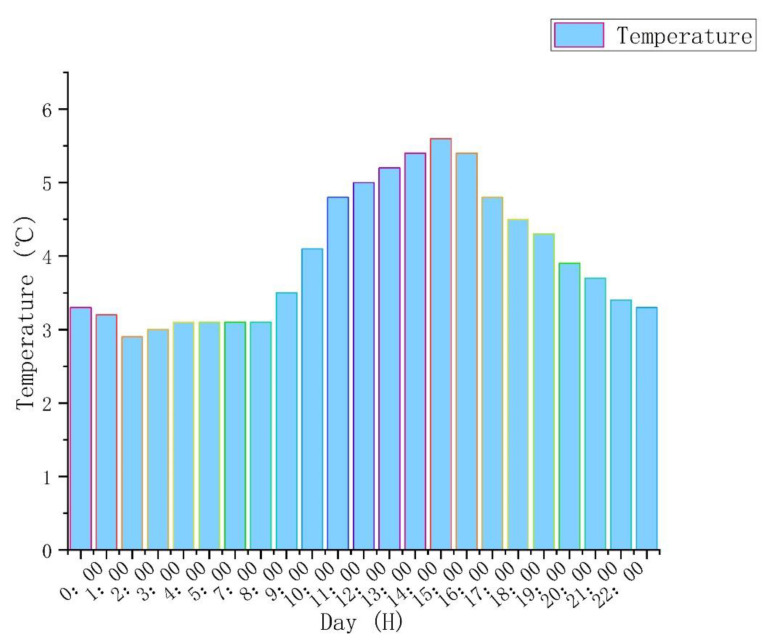
Change in outdoor temperature.

**Figure 5 ijerph-18-12600-f005:**
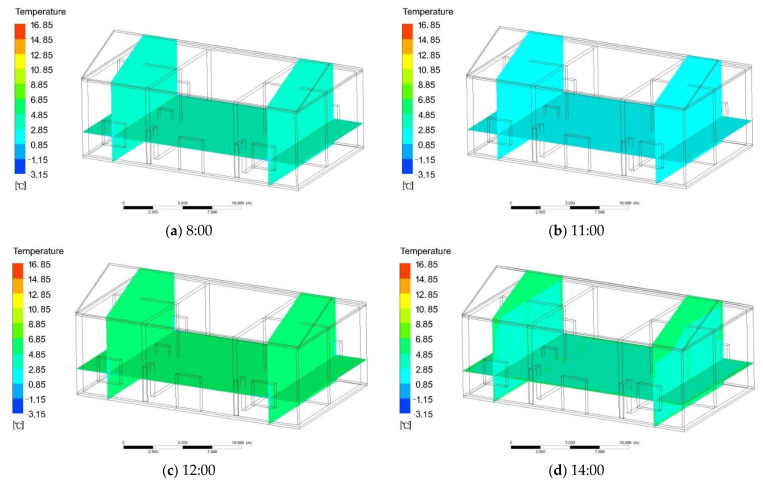
(**a**,**b**) Change in indoor temperature from 8:00 to 12:00; (**c**,**d**) change in indoor temperature from 12:00 to 14:00; (**e**) change in indoor temperature from 14:00 to 17:00; (**f**) change in indoor temperature from 17:00 to 18:00.

**Figure 6 ijerph-18-12600-f006:**
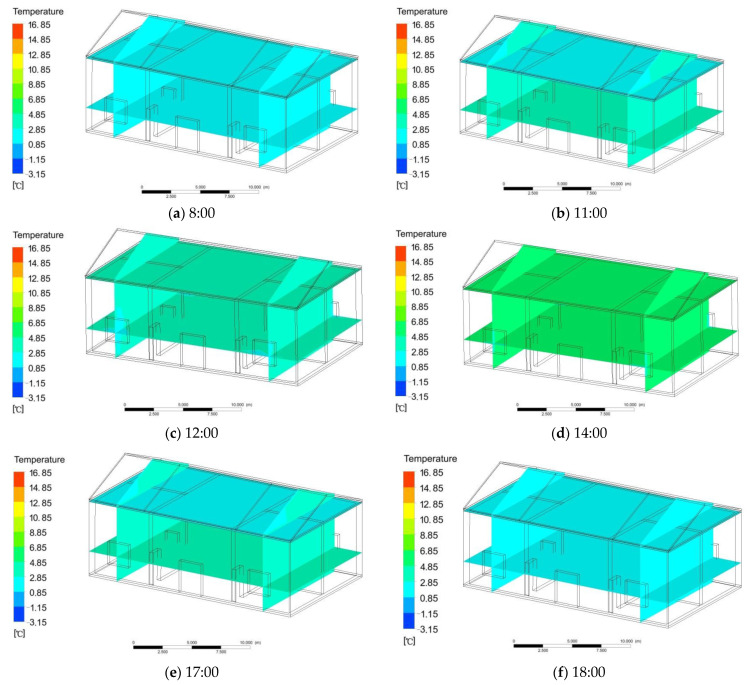
(**a**,**b**) Change in indoor temperature from 8:00 to 12:00; (**c**,**d**) change in indoor temperature from 12:00 to 14:00; (**e**) change in indoor temperature from 14:00 to 17:00; (**f**) change in indoor temperature from 17:00 to 18:00.

**Figure 7 ijerph-18-12600-f007:**
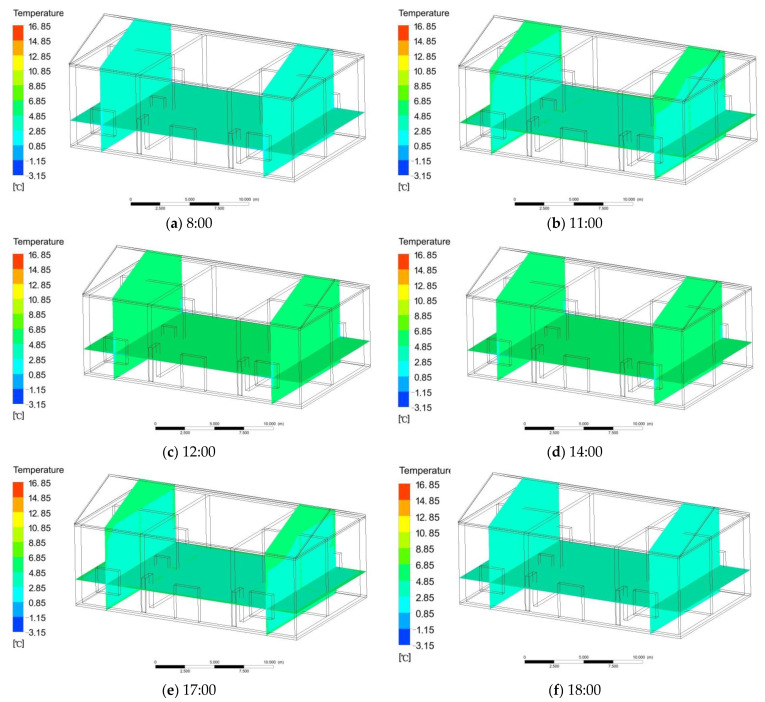
(**a**,**b**) Change in indoor temperature from 8:00 to 12:00; (**c**,**d**) change in indoor temperature from 12:00 to 14:00; (**e**) change in indoor temperature from 14:00 to 17:00; (**f**) change in indoor temperature from 17:00 to 18:00.

**Figure 8 ijerph-18-12600-f008:**
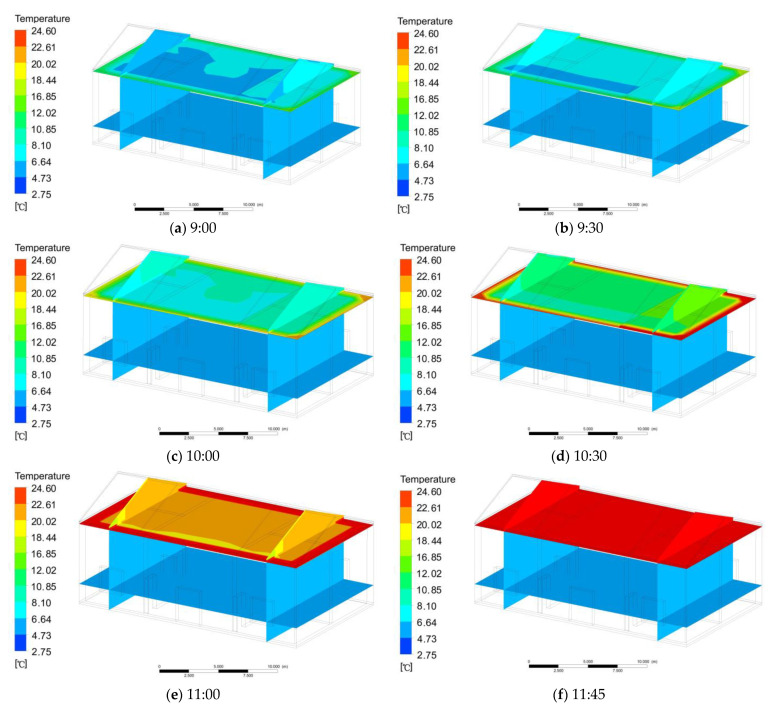
(**a**) Change in indoor temperature at 9:00; (**b**) change in indoor temperature at 9:30; (**c**) change in indoor temperature at 10:00; (**d**) change in indoor temperature at 10:30; (**e**) change in indoor temperature at 11:00; (**f**) change in indoor temperature at 11:45.

**Figure 9 ijerph-18-12600-f009:**
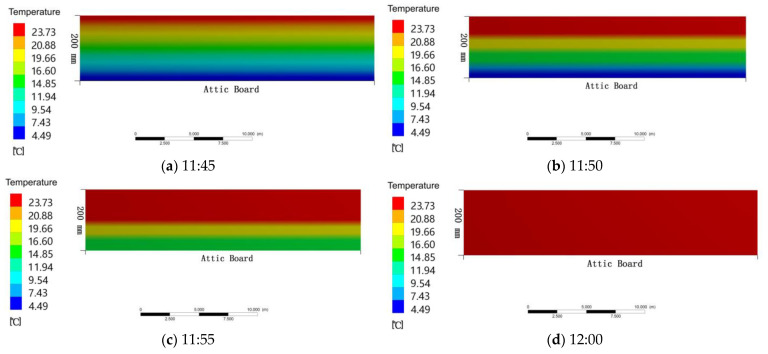
(**a**) Temperature change of the attic floor at 11:45; (**b**) temperature change of the attic floor at 11:50; (**c**) temperature change of the attic floor at 11:55; (**d**) temperature change of the attic floor at 12:00.

**Figure 10 ijerph-18-12600-f010:**
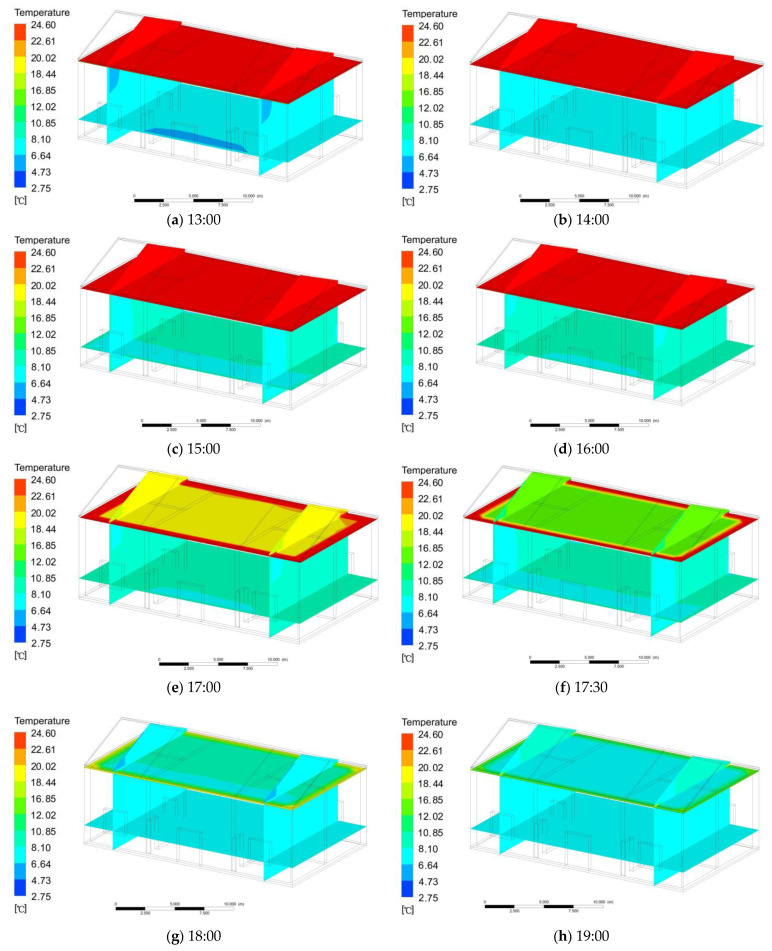
(**a**) Indoor temperature at 13:00; (**b**) indoor temperature at 14:00; (**c**) indoor temperature at 15:00; (**d**) indoor temperature at 16:00; (**e**) indoor temperature at 17:00; (**f**) indoor temperature at 17:30; (**g**) indoor temperature at 18:00; (**h**) indoor temperature at 19:00.

**Figure 11 ijerph-18-12600-f011:**
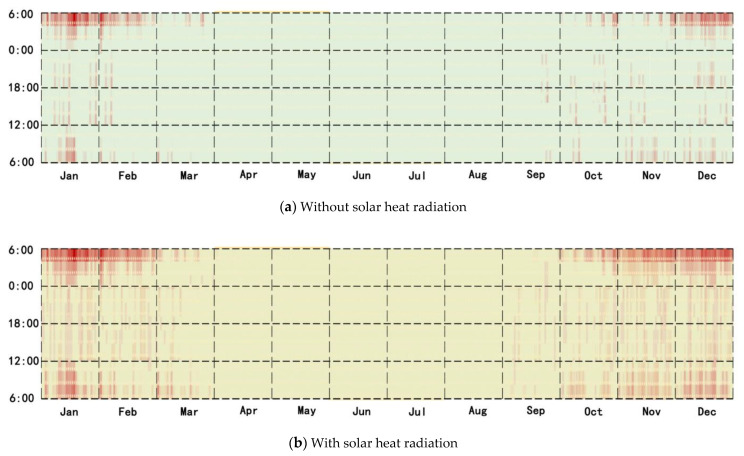
(**a**) Heating time of the new system without solar heat radiation; (**b**) heating time of the new system.

**Figure 12 ijerph-18-12600-f012:**
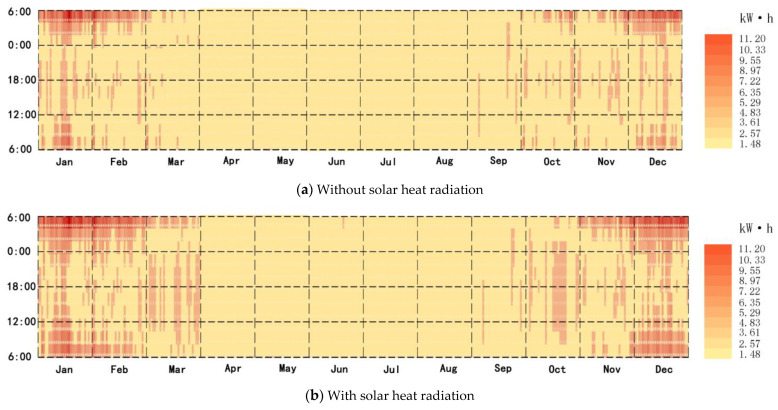
(**a**) Energy-saving situation of using the new system without solar heat radiation; (**b**) energy-saving situation of using the new system with solar thermal radiation.

**Figure 13 ijerph-18-12600-f013:**
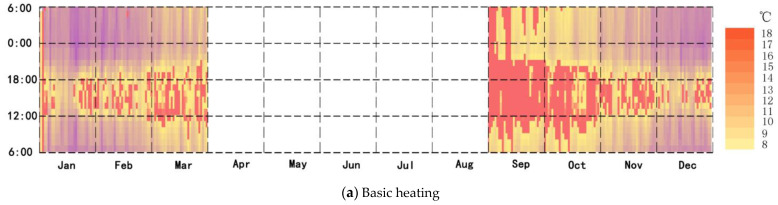
(**a**) Thermal comfort of the basic heating; (**b**) thermal comfort of the new system without solar heat radiation; (**c**) thermal comfort of the new system with solar heat radiation.

**Table 1 ijerph-18-12600-t001:** Thermal properties of some PCMs.

Material	Melting Temperature (°C)	Heat of Fusion (kJ/kg)
HDPE	120–135	300
MgC_l2_·6H_2_O	117	168.6
Paraffin wax	64	173.6
Polyglycol E6000	66	190
Biphenyl	71	119.2
Naphthalene	80	147.7
Palmitic acid	64	185.4
Stearic acid	69	202.5

**Table 2 ijerph-18-12600-t002:** Change in indoor temperature.

DAY	Indoor Temperature with Closed Heat Shield (°C)	Indoor Temperature with Open Heat Shield (°C)
1	5.9	3.7
2	6.3	3.9
2	5.9	3.6
3	6.4	3.6
4	6.3	3.9
5	6.2	3.9
6	6.5	3.8
7	5.9	4
8	6.3	3.7
9	6.3	3.9
10	5.9	3.7
Average temperature	6.2	2.8

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
