# Peer review of "Study on the Roof Solar Heating Storage System of Traditional Residences in Southern Shaanxi, China"

_ijerph, 2021, doi:10.3390/ijerph182312600_

Round 1

Reviewer 1 Report

The paper describes model simulations of a roof solar collector system with a thermal storage plate towards the rooms below in traditional houses in southern Shaanxi province of China and concludes that this system can increase the temperatures in the rooms and reduce the use of other traditional heating sources and thus reducing the carbon emission and pollution.

The paper does a good job of showing that the suggested passive application can improve the thermal indoor climate and thus greatly improve the living conditions without any need of maintenance. 

The temperatures are very low and thus way below any reasonable phase shifting temperatures of the high-density polyethylene board that is placed on the attic board. Thus the emphasis on this aspects may not be relevant for the overall analysis. It may be cheaper to just add more materials than to use a more costly material to achieve the desired effect anyway as overheating does not seem to be a large problem in the investigated region. 

  • In Line 245 - This is an unfinished sentence and i expect there should be a reasoning for the choice of material. I can however not find it and as stated above - I dont there is a reasoning for phase change. Also the suggested summer operation does not imply the use of phase change to absorb summer heat but rather to ventilate the heat away.

I look forward to see a demonstration project utilising the suggested approach and to investigate the practicalities of the system. 

Reviewer 2 Report

The title is not very clear, and needs to be improved

What does a solar heating system of thermal storage roof pool of traditional residence … mean???

The English written of the abstract must be improved, with a more coherent way of writing.

In these lines 167-169:

“The roof attic board is set as a solar panel thermal storage material to absorb the heat radiation of the sun and increase the overall indoor temperature through thermal convection and thermal radiation.”

I can guess what this sentence wants to say, but its readability needs to be improved.

In this paragraph: lines 58-62:

“This research puts forward a new passive heating design, which combines the roof of the building with renewable energy and increases the indoor temperature through the thermal capacity of the floor, that is, the design of the heat storage roof pool solar heating system. Its purpose is to reduce energy consumption, improve energy utilization, and reduce carbon emissions.”

  • I believe the gap in knowledge and possible limitations should be clearer written before this paragraph, and it should be highlighted how your research is going to close this gap. Then further relevant background study can be reviewed in section 1.2.

Line 64: “Regarding the research on building energy saving, some scholars have proposed the use of new energy to optimize the indoor temperature to save energy [10], ”

“, some scholars have proposed the use of new energy” is not clear at all! Some researchers have proposed the use of new energy?

Provide proper references in “Simulation using ANSYS software proved the effectiveness of 188 the new system for heating. The specific goals are as follows:” revise the whole manuscript for missed references.

Revise the title/format of Figure (a) Xian He Kou Village. What is the scientific meaning of this figure?

This manuscript needs major revision. The topic of study itself could be of interest, however, the way that this research is written must be extensively improved. The article needs to be written in a more cohesive way, try to avoid vague writing.

The English language and style of this manuscript must be checked with a professional scientific / engineering manuscript reviewer and maybe revised by a native / professional engineer with experience in this field.

Thank you.

Round 2

Reviewer 2 Report

Thanks to authors to extensively revise their manuscript. I think now this research can be published in the journal. However, I highly recommend the authors to address these two issues mentioned below:

Please check Figure 1, the earth view figure overlaps the graph in the pdf file. Maybe in word or latex the format is fine but what I see in the PDF version is not correct.

In Figure 3, please add left-y axis Lable, if it's unitless just use [-], and change usage frequency to “Usage frequency” please. It’s a typo. 

Thank you and I wish you all the best in your future endeavours.
